# Magnetized fast isochoric laser heating for efficient creation of ultra-high-energy-density states

Shohei Sakata [1], Seungho Lee [1], Hiroki Morita[1], Tomoyuki Johzaki [2], Hiroshi Sawada [1,3], Yuki Iwasa[1], Kazuki Matsuo [1], King Fai Farley Law [1], Akira Yao[1], Masayasu Hata[1], Atsushi Sunahara [4,9], Sadaoki Kojima [1,10], Yuki Abe [1], Hidetaka Kishimoto[1], Aneez Syuhada[1], Takashi Shiroto [5], Alessio Morace[1], Akifumi Yogo[1], Natsumi Iwata[1], Mitsuo Nakai[1], Hitoshi Sakagami[6], Tetsuo Ozaki[6], Kohei Yamanoi[1], Takayoshi Norimatsu[1], Yoshiki Nakata [1], Shigeki Tokita[1], Noriaki Miyanaga[1], Junji Kawanaka [1], Hiroyuki Shiraga[1], Kunioki Mima[1,7], Hiroaki Nishimura[1], Mathieu Bailly-Grandvaux [8], João Jorge Santos [8], Hideo Nagatomo[1], Hiroshi Azechi[1], Ryosuke Kodama[1], Yasunobu Arikawa[1], Yasuhiko Sentoku[1] & Shinsuke Fujioka [1]

Fast isochoric heating of a pre-compressed plasma core with a high-intensity short-pulse laser is an attractive and alternative approach to create ultra-high-energy-density states like those found in inertial confinement fusion (ICF) ignition sparks. Laser-produced relativistic electron beam (REB) deposits a part of kinetic energy in the core, and then the heated region becomes the hot spark to trigger the ignition. However, due to the inherent large angular spread of the produced REB, only a small portion of the REB collides with the core. Here, we demonstrate a factor-of-two enhancement of laser-to-core energy coupling with the magnetized fast isochoric heating. The method employs a magnetic field of hundreds of Tesla that is applied to the transport region from the REB generation zone to the core which results in guiding the REB along the magnetic field lines to the core. This scheme may provide more efficient energy coupling compared to the conventional ICF scheme.

[1] Institute of Laser Engineering, Osaka University, 02-06 Yamada-Oka, Suita, Osaka 565-0871, Japan. [2] Department of Mechanical Systems Engineering, Hiroshima University, Higashi-Hiroshima, Hiroshima 739-8527, Japan. [3] Department of Physics, University of Nevada Reno, Reno, Nevada 98557, USA. [4] Institute for Laser Technology, 1-8-4 Utsubo-honmachi, Nishi-ku Osaka, Osaka 550-0004, Japan. [5] Department of Aerospace Engineering, Tohoku University, Sendai, Miyagi 980-8579, Japan. [6] National Institute for Fusion Science, National Institutes of Natural Sciences, 322-6 Oroshi, Toki, Gifu 509-5292, Japan. [7] The Graduate School for the Creation of New Photonics Industries, 1955-1, Kurematsu, Nishi-ku, Hamamatsu, Shizuoka 431-1202, Japan. [8] University of Bordeaux, CNRS, CEA, CELIA (Centre Lasers Intenses et Applications), UMR 5107, F-33405 Talence, France. [9] Present address: Center of Materials Under eXtreme Environment, Purdue University, 500 Central Drive, West Lafayette, Shizuoka Indiana 47907, USA. [10] Present address: Advanced Research Center for Beam Science, Institute for Chemical Research, Kyoto University, Gokasho, Uji Kyoto 611-0011, Japan. Correspondence and requests for materials should be addressed to S.F. (email: sfujioka@ile.osaka-u.ac.jp)

A large energy, high-power, high-intensity laser can create a large volume of ultra-high-energy-density plasma for particle accelerators[1], planetary science[2,3], astrophysics[4,5], and nuclear physics[6,7]. Inertial confinement fusion (ICF) is an ultimate application of such extreme plasmas[8–10].

A few millimeter-scale spherical capsule used in the ICF contains a deuterium–tritium (DT) fusion fuel ice layer. In the laser indirect-drive approach, the capsule surface is irradiated by X-rays in a high-Z metal enclosure (hohlraum). The X-rays drive a sequence of converging shock waves and compressing fusion fuel > 1000 times solid density. Adiabatic compression heats up the DT gas initially filling the capsule interior, which becomes the ignition spark at the final stage of the compression.

The scientific breakeven, where the energy released by fusion reaction exceeds the energy contained in the compressed fusion fuel, was achieved in National Ignition Facility (NIF)[11]; however, the pathway to ICF ignition is still unclear. One of the crucial problems of the current central ignition scheme is the hot spark mixing with the cold dense fuel because of significant growth of hydrodynamic instabilities during the compression.

Fast isochoric heating, also known as fast ignition[12], of a pre-compressed core, was proposed as an alternative approach to the ICF ignition that avoids the ignition quench caused by the mixing because the hot spark is generated not by the adiabatic compression but by the external energy injection whose timescale is shorter than the hydrodynamic timescale (<100 ps). Relativistic intensity laser pulses ($I_L \lambda_L^2 > 1.37 \times 10^{18}$ W μm$^2$ cm$^{-2}$) efficiently produce relativistic electron beams (REB) via laser–plasma interactions[13–15]; here, $I_L$ and $\lambda_L$ are laser intensity [W cm$^{-2}$] and wavelength [μm], respectively. The REB travels in a plasma from its generation zone to the core. A part of the REB kinetic energy is deposited into the core, and then the heated region becomes the hot spark to trigger the fusion ignition.

Several breakthroughs were described in past articles on fast heating research: the invention of the cone-in-shell target[16], high areal density core formation with the cone-in-shell target[17], and efficient laser-to-core coupling after reduction of pre-plasma filling in the cone[18]. Our achievement, namely enhanced laser-to-core energy coupling with the magnetized fast isochoric heating enabled by an application of external kilo-Tesla-level magnetic field, is also a milestone as a demonstration of the efficient energy coupling, as a means to reduce the large radial spread of the REB. This is considered essential to secure scalability of the fast heating to the ignition[19].

Three critical problems have been recognized as obstacles to the efficient fast heating with the REB. The first problem is that the REB generated in a long-scale-length pre-plasma filled in the cone becomes too energetic to heat the core, the second one is that a part of the REB is scattered and absorbed in a high-Z cone tip[20]. The third one is that the REB has a large divergence of 100° as a typical full-angle, so that only a small fraction of the diverged REB can collide with the core[21].

The long-scale-length pre-plasma filling the cone is produced by prepulse and foot pulses of the heating laser and also by the cone breakup due to high pressure of a plasma surrounding the cone. The heating efficiency of 7% was attained at OMEGA laser facility with an ignition-scale large areal density ($\rho R \sim 0.3$ g cm$^{-2}$) core[17] after reducing the pre-plasma formation in the cone[18]; however, the aforementioned second and third critical problems still remain to be resolved.

Here, we have introduced two experimental techniques. A solid ball target is used for making an open-tip cone utilizable along with the plasma compression. The solid ball compression does not generate shocks and rarefactions traveling ahead of the dense shell; therefore, a closed tip is not required for preventing a hot

plasma from filling inside the cone. In addition, a relatively cold and dense core can be produced stably by using the solid ball target. The cold core enables us to visualize REB transport region[18] in a dense core by using monochromatic Cu-$K_\alpha$ imaging technique without a significant energy shift of the Cu-$K_\alpha$ photon energy due to ionization of Cu atoms. The use of a solid ball target has another benefit for producing a moderate guiding field as discussed later.

Another technique is to use a laser-driven capacitor–coil target[22] to generate a strong magnetic field. The strength of the magnetic field was measured on GEKKO-XII[23,24], LULI2000[25], Shengguang-II[26], and OMEGA-EP laser facilities[27,28]. As discussed in the Supplementary Discussion and summarized in Supplementary Table 1, experimental results revealed that a 600–700-T magnetic field was generated by using a tightly focused kilo-joule and nano-second infrared ($\lambda_L = 1.053$ μm) laser beam.

Application of external magnetic fields to the path of a REB is expected to guide the diverging REB to the dense core[19]. For example, the gyroradius of a 1-MeV electron under the influence of a 1-kT magnetic field is 5 μm, which is smaller than a typical core radius (20 μm); therefore, a kilo-Tesla-level magnetic field is sufficient to guide the REB to the core. The guidance of the REB by the laser-produced external magnetic field has already been demonstrated experimentally in an uncompressed-planar geometry at LULI2000 facility (Ecole Polytechnique, France)[29].

In this study, we have demonstrated the enhancement of the laser-to-core energy coupling with the magnetized fast isochoric heating. The maximum coupling of 7.7 ± 1.2% was achieved even with a relatively small radial areal density core ($\rho R = 0.08$ g cm$^{-2}$), which could be higher than that has been achieved by the fast isochoric heating without applying a magnetic field[18]. This value is fairly consistent with the simple evaluation, which reveals that a higher areal density core leads to higher laser-to-core coupling.

## Results

**Laser-to-core coupling measurement.** The laser-to-core energy couplings were experimentally measured by systematically varying the experimental conditions: heating laser energy, injection timing of the heating laser, application or not of the external magnetic field, and open- or closed-tip cones.

Table 1 summarizes laser-to-core coupling efficiency obtained in this experiment. The data are separated into three groups by the rules according to the experimental conditions, where the external magnetic field was applied or not, and the cone-tip was open or closed. The data are sorted by the laser-to-core coupling efficiency in each group.

The coupling was calculated from the absolute number of Cu-$K_\alpha$ X-ray (8.05 keV) photons emitted from Cu atoms contained in the laser-compressed hydrocarbon core. Cross-sections of electron-impact $K$-shell ionization have a similar dependence on electron energy as collisional energy loss. The two are essentially the same process but with a different threshold energy. Collisional deposition of REB energy (J) in a Cu-contained core can be obtained from the number of Cu-$K_\alpha$ photons (photons/sr) emitted from the core[18] with a correlation factor. Details of the correlation factor derivation are described in the Methods section.

Figure 1a shows an experimental layout. The experiment was conducted on the GEKKO-LFEX laser facility at the Institute of Laser Engineering, Osaka University. The fusion fuel surrogate was made of a 200-μm-diameter solid Cu(II) oleate solid ball [Cu (C$_{17}$H$_{33}$COO)$_2$][30], whose surface was coated with a 25-μm-thick polyvinyl alcohol (PVA) layer to prevent the Cu atoms from being ionized directly by the compression laser beams. The Cu(II) oleate ball contains 9.7% Cu atoms in weight for visualization of

**Table 1 Summary of laser-to-core coupling efficiencies**

| Shot ID | Cone tip condition | Heating energy [J] | Compression energy [J] | B-generation energy [J] | Heating timing [ns] | Cu-Kα number [photons sr−1] | Coupling efficiency [%] |
|---|---|---|---|---|---|---|---|
| 40545 | Open | 899 | 1422 | N/A | 0.42 | $5.58 \times 10^{11}$ | 2.9 ± 0.5 |
| 40541 | Open | 683 | 1428 | N/A | 0.65 | $5.53 \times 10^{11}$ | 3.9 ± 0.6 |
| 40558 | Open | 1516 | 1386 | 1761 | 0.4 | $1.19 \times 10^{12}$ | 3.1 ± 0.5 |
| 40556 | Open | 1016 | 1332 | 1698 | 0.61 | $1.02 \times 10^{12}$ | 4.3 ± 0.7 |
| 40547 | Open | 1100 | 1530 | 1824 | 0.38 | $1.28 \times 10^{12}$ | 5.5 ± 0.9 |
| 40549 | Open | 668 | 1548 | 1794 | 0.37 | $7.29 \times 10^{11}$ | 5.8 ± 0.9 |
| 40543 | Open | 625 | 1494 | 1842 | 0.72 | $9.32 \times 10^{11}$ | 7.7 ± 1.2 |
| 40560 | Closed | 1523 | 1404 | 1794 | 0.38 | $8.23 \times 10^{11}$ | 2.5 ± 0.4 |
| 40562 | Closed | 1378 | 1374 | 1725 | 0.65 | $7.96 \times 10^{11}$ | 2.7 ± 0.4 |

The data are separated into three groups by the rules according to the experimental conditions, where the external magnetic field was applied or not, and the cone-tip was open or closed. The data are sorted by the laser-to-core coupling efficiency in each group.

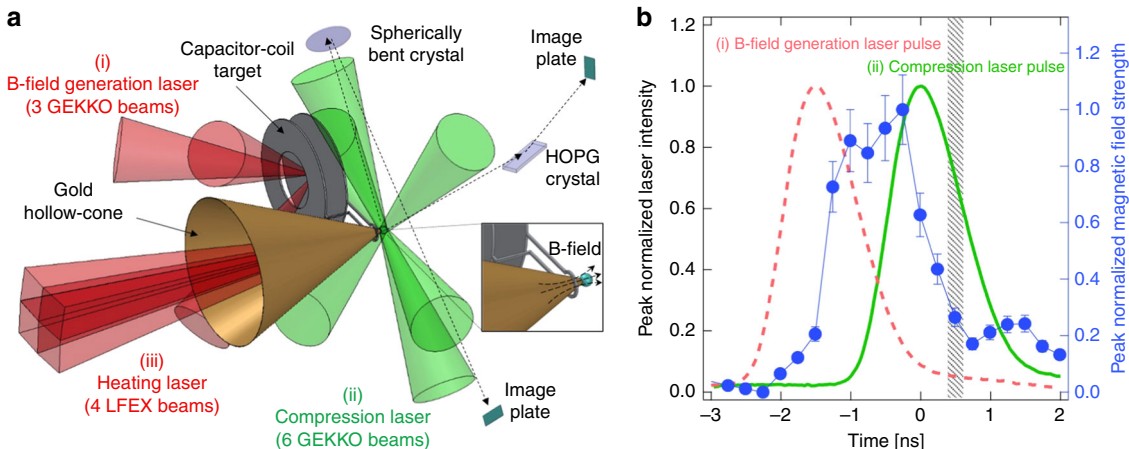

**Fig. 1** Experimental layout for the laser-to-core energy coupling measurement. **a** A schematic drawing of the experimental layout for the magnetized fast isochoric heating. Three and six of GEKKO-XII laser beams were used for generation of the magnetic field using the capacitor–coil target and compression of the solid ball, respectively. Four LFEX laser beams were irradiated on the tip of the cone to produce a REB. The X-ray image using a spherically bent crystal and the X-ray spectrometer using a HOPG crystal were used for visualization of the REB transport and measurement of the laser-to-core energy coupling, respectively. **b** Timing chart of the magnetic-field-generation laser (pink broken line), fuel compression laser (green solid line), and laser-produced magnetic field (blue circular marks) pulses. The error bars show the uncertainty of magnetic field measurement described in ref. [24]. The hatching area indicates the injection timing of the heating laser

the relativistic electron beam (REB) transport in the core and for measurement of the laser-to-core energy coupling (9.7% is a little bit smaller than the expected value (10.1%) because of a contamination inclusion in the Cu(II) oleate). The fuel surrogate was attached to a Au cone, whose open angle, wall thickness, and tip diameter were 45°, 7 μm, and 100 μm, respectively. The outer surface of the Au cone was coated with a 50-μm-thick PVA layer to delay the cone breakup time[31]. Open-tip or closed-tip Au cones were used in the experiments; the tips of the closed cones were covered with a 7-μm-thick Au layer.

The solid ball was compressed by six of GEKKO-XII laser beams arranged in a quasi-cylindrical geometry, whose wave-length, pulse shape, pulse duration, and energy were 0.526 μm, Gaussian, 1.3 ns full-width at half-maximum (FWHM), and 240 ± 15 J per beam, respectively. The center of the nickel-made coil having a 500-μm diameter was located 230 μm from the center of the ball to apply a strong magnetic field near the REB generation zone and the solid ball. The first disk of the laser-driven capacitor was irradiated through the hole of the second disk by three tightly focused GEKKO-XII laser beams, whose wavelength, pulse shape, pulse duration, and energy were 1.053 μm, Gaussian, 1.3 ns

(FWHM), and 600 ± 20 J per beam, respectively, yielding $7 \times 10^{15}$ W cm$^{-2}$ of the peak intensity.

The tip of the cone was irradiated to produce a REB by four LFEX laser beams, whose wavelength, pulse shape, and pulse duration were 1.053 μm, Gaussian, and 1.8 ± 0.3 ps (FWHM), respectively. The total energy of the four LFEX beams on the tip was varied from 630 to 1520 J. The focal spot diameter was 50 μm (FWHM) containing 30% of the total energy, yielding an intensity of $1.3 \times 10^{19}$ W cm$^{-2}$ at the maximum energy shot.

The laser energy and injection timing of the heating laser are also summarized in Table 1. The injection timings were measured with an X-ray streak camera with an accuracy of ± 0.02 ns. Figure 1b shows a time chart of the magnetic-field-generation laser, compression laser, and laser-produced magnetic field pulses. The time origin ($t = 0$ ns) is defined hereafter as the peak of the compression laser pulse. The peak of the magnetic-field-generation laser pulse was set at $t = -1.5$ ns; therefore, the magnetic field strength reaches its maximum value before the compression beam irradiation. The heating lasers (four LFEX beams) were injected around the maximum compression timing $t = 0.38–0.72$ ns shown as the hatching area in Fig. 1b.

Copper $K_\alpha$ X-rays (8.05 keV) were imaged using a spherically bent quartz (2131) crystal to visualize the transport of the REB in the pre-compressed core from the direction perpendicular to the LFEX incident axis. The magnification, spatial resolution, and spectral bandwidth were 20, 13 μm (FWHM), and 5 eV (FWHM), respectively.

The X-ray spectrometer using a planar highly oriented pyrolytic graphite (HOPG), was installed at 40° from the LFEX incident axis to measure the absolute Cu-$K_\alpha$ yield. The absolute integral reflectance of the HOPG was measured using an X-ray diffractometer. The HOPG has ±16% of spatial nonuniformity of the integral reflectance. The spectral resolution of the spectrometer was 17.9 eV (FWHM).

Figure 2 shows an example of Cu-$K_\alpha$ spectra. Cu-$K_\alpha$ X-ray yield produced during the compression process (green dotted line) was negligibly weak compared with those produced with the heating lasers (red solid and black dashed lines). The green dotted and red solid lines are spectra obtained with application of the external magnetic field. The vertical error bar corresponds to spatial nonuniformity of the integral reflectance, and the horizontal error bar is equal to the spectral resolution. The Cu-$K_\alpha$ photon yields were integrated within the energy range of 8.0–8.1 keV after background subtraction.

**Initial magnetic field profile calculation.** An externally applied magnetic field penetrates diffusively into the target from the outside. The diffusion timescale is determined by the electrical conductivity and geometry of the targets. Based on a previous study, the externally applied magnetic field is guaranteed to penetrate rapidly into an insulator hydrocarbon[29]; however, the diffusion dynamics of the magnetic field into the gold cone remained unclear.

The magnetic diffusion time ($t_{dif}$) is expressed as $t_{dif} = \mu_0 \sigma aL/2$ for a cylindrical tube that is topologically similar to a cone, where $\mu_0$, $\sigma$, $a$, and $L$ are the permeability, electrical conductivity, radius, and diffusion layer thickness of the tube, respectively. The temporal change in magnetic field strength drives an inductive current in the gold cone, and the current ohmically heats the gold. The electron conductivity of the gold depends on its temperature.

For a 7-μm-thick gold cone wall at 50-μm radius cone tip, the diffusion times are $t_{dif} = 18$ ns, 860 ps, and 420 ps at room temperature ($\sigma = 4 \times 10^7$ S m$^{-1}$), 0.1 eV ($\sigma = 2 \times 10^6$ S m$^{-1}$), and 1 eV ($\sigma = 1 \times 10^6$ S m$^{-1}$), respectively[32]. The small temperature increment helps to rapidly soak the cone in the magnetic field.

For more detailed estimation, we developed an electro-magneto dynamics simulation code with consideration of the inductive heating and the temperature dependence of conductivity to calculate the spatial and temporal distribution of a magnetic field in the cone-in-ball target[33]. A current of 250 kA was generated with a capacitor–coil target driven by one GEKKO-XII beam as measured using proton radiography[24]. The current of 250 kA was used in the following analysis, although three GEKKO-XII beams were used in this integrated experiment.

Figure 3 shows the two-dimensional profile of the magnetic field calculated at the maximum magnetic field strength timing generated using a 250-kA Gaussian current pulse with a 1-ns (FWHM) pulse width. The diffused magnetic field profiles were calculated using a constant electrical conductivity $\sigma = 4 \times 10^7$ of a cold gold (Fig. 3a) or temperature-dependence conductivity of gold (Fig. 3b). The magnetic field strengths at the tip are $B_{int} = 96$ and 335 T, respectively. Temporal changes of the magnetic field profile are discussed in Supplementary Methods and are shown in Supplementary Fig. 2.

In this study, the magnetic field lines are bent due to magnetic field compression associated with the plasma compression. If a magnetic mirror is formed in the transport region, reflection of the REB by the magnetic mirror could degrade the laser-to-core coupling. The magnetic field strength at the ball center is $B_0 = 225$ T before the compression, as shown in Fig. 3b. The ball radius at the maximum compression timing was about $r_{min} = 50$ μm, as shown in Fig. 4b. The magnetic field strength at the maximum compression timing ($B_{max}$) can be estimated with $B_{max} = B_0 (r_0/r_{min})^{2(1-1/Re_m)}$, where $r_0 = 125$ μm and $Re_m \sim 2$[31,34] are initial radius of the ball and magnetic Reynolds number, respectively, which gives $B_{max} = 560$ T. Therefore, the mirror

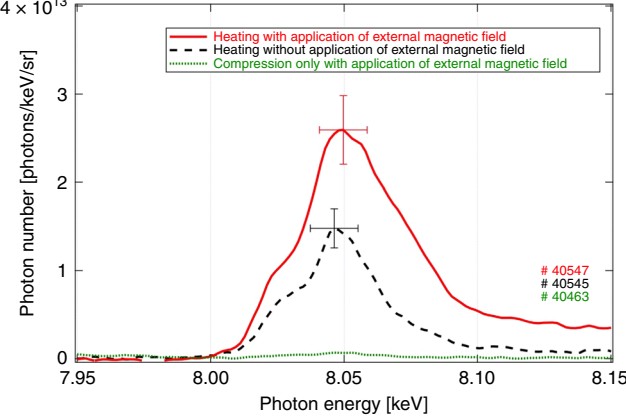

**Fig. 2** An example of Cu-$K_\alpha$ spectra peaked at 8.05 keV emitted from the compressed plasma core. Red solid, black dashed, and green dotted lines are, respectively, spectra obtained by heating with application of an external magnetic field, heating without the application, and only fuel compression with application of an external magnetic field. The vertical error bar corresponds to 16% of spatial nonuniformity of the integral reflectance of the HOPG. The horizontal error bar of 17.9 eV is the spectral resolution of the spectrometer. The Cu-$K_\alpha$ photon yields were integrated within the energy range of 8.0–8.1 keV

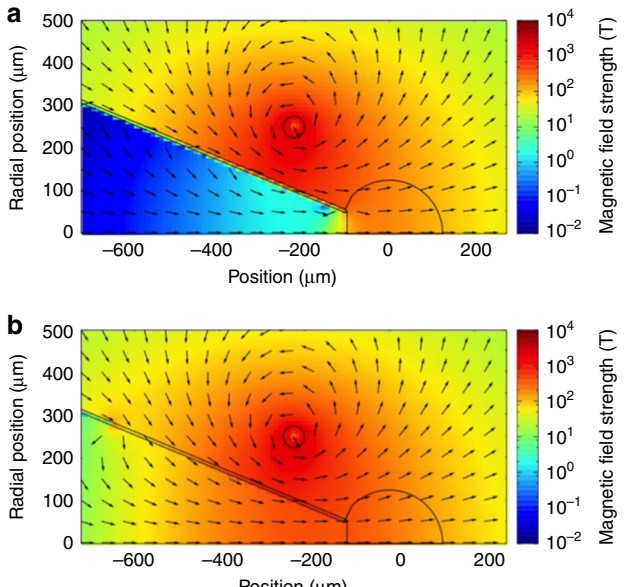

**Fig. 3** Two-dimensional profiles of the magnetic field calculated with different electrical conductivities. The magnetic field diffusion at the peak timing of current flow in the coil was calculated using (**a**) the constant electrical conductivity [$\sigma = 4 \times 10^7$ S m$^{-1}$] and (**b**) with consideration of temporal changes in temperature and conductivity of the gold cone due to the inductive heating

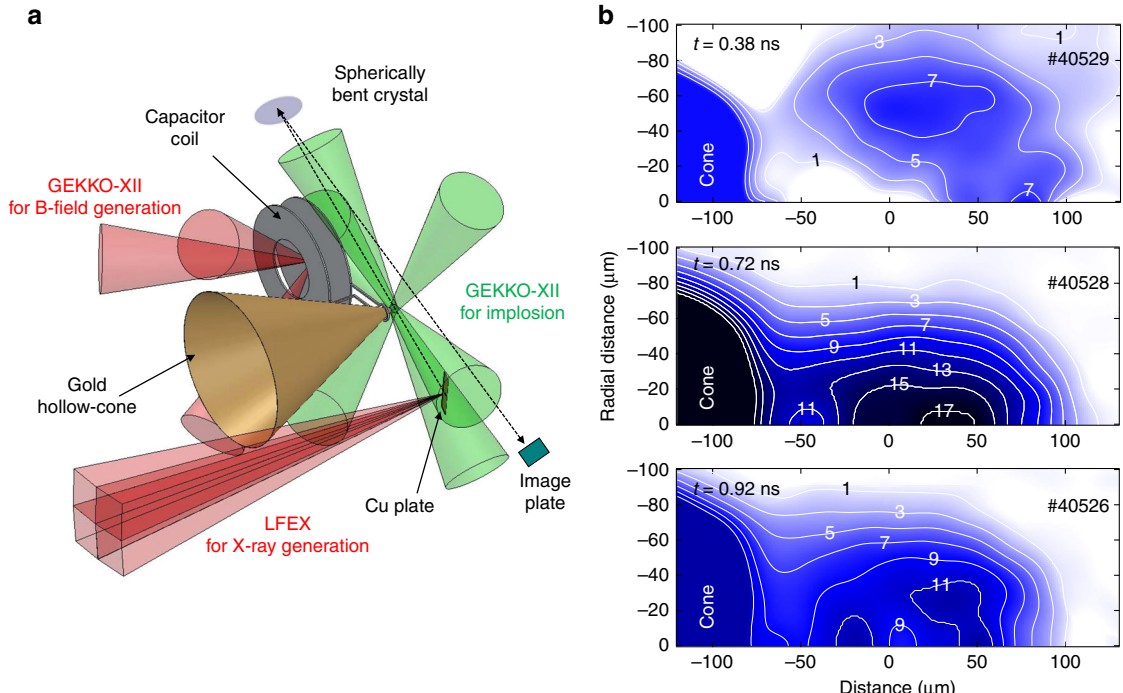

**Fig. 4** The core density profiles of the pre-compressed core at three different timings. **a** Experimental layout of the pre-compressed core density measurement experiment. The LFEX laser was used to generate a Cu-$K_\alpha$ backlight flash. **b** Density profiles measured at $t = 0.38$, 0.72, and 0.92 ns after the peak of the compression beam pulse. The numbers on the contour lines represent mass density (g cm$^{-3}$)

ratio along the REB path, the ratio of magnetic field strengths at the peak and the generation zone, is $R_m = B_{max}/B_{int} = 1.7$.

The REB transport was simulated in the mirror geometry with several mirror ratios ($R_m$) from 0 to 20[35], and it was determined that a moderate mirror ($R_m < 10$) can focus on the REB without a significant loss caused by the mirror effect. This mirror ratio $R_m = 1.7$ is small enough to guide the REB efficiently to the core in this system with avoiding the mirror effect.

**Two-dimensional density profile measurement**. Flash X-ray backlighting with a monochromatic imager[17,36–38] was used to measure two-dimensional density profiles of pre-compressed Cu (II) oleate solid balls under an external magnetic field.

The experimental layout is shown in Fig. 4a. The X-ray shadows of compressed cores were recorded using imaging plates with the same spherically bent quartz crystal used in the laser-to-core coupling experiment. The solid ball specifications and the laser parameters of the compression and magnetic-field-generation beams were also identical to those used in the laser-to-core coupling measurement. The LFEX laser was used for flash Cu-$K_\alpha$ X-ray backlight generation in this experiment. In order to generate a large-format backlight, the LFEX laser was defocused to produce a 350-μm-diameter spot on a 20-μm-thick Cu foil at 3 mm behind the solid ball along the line of sight of the crystal imager.

An X-ray shadow is converted to an X-ray transmittance profile by interpolating the two-dimensional backlight X-ray intensity profile within the core region from the outside of the core region. The areal density of the pre-compressed core was calculated from the X-ray transmittance profile with a calculated opacity of 100 eV Cu(II) oleate for 8.05-keV X-rays[39]. A two-dimensional density profile of the core was obtained after applying an inverse Abel transformation to the areal density profile, assuming rotational symmetry of the core along the cone axis.

Figure 4b shows the core density profiles at $t = +0.38$, $+0.72$, and $+0.92$ ns. The numbers on the contour lines represent mass density (g cm$^{-3}$). The converging shock wave was still traveling to the center of the ball at $t = +0.38$ ns, maximum compression was reached at around $t = +0.72$ ns, and the core had already begun to disassemble at $t = +0.92$ ns. The areal mass densities ($\rho L$) and average mass density ($\rho$) of the core along the REB path length ($L$) were, respectively, $\rho L = 0.08$ g cm$^{-2}$ and $\rho = 5.7$ g cm$^{-3}$ at $t = +0.38$ ns, $\rho L = 0.16$ g cm$^{-2}$, and $\rho = 11.3$ g cm$^{-3}$ at $t = +0.72$ ns. These values were used in the calculation of the correlation factor described in the Methods section.

## Discussion

Figure 5 shows the dependence of the measured coupling efficiency on the heating laser intensity (bottom axis) and energy (top axis). Solid and open marks represent the couplings of two injection-timing groups $t = 0.61–0.72$ ns and $t = 0.37–0.42$ ns. The solid and dashed lines are fitted, as an eye guide, to the couplings measured with or without application of the external magnetic field neglecting the injection timing difference.

It is not possible to identify the exact area of the HOPG that diffracts the detected Cu-$K_\alpha$ X-rays. Therefore, we considered the ±16% of nonuniformity of the integral reflectance over its full surface to impact the uncertainties in evaluations of the deposited energy and the laser-to-core coupling. Nonetheless, it is important to refer here that the HOPG alignment in respect to the targets was not changed during these measurements, and the detected $K_\alpha$ photons were always reflected from the same area in the crystal. This fact means that the true coupling points do not scatter randomly within the error bars but are at certain points keeping relations between other points within the error bars.

The figure reveals clearly that the REB guiding by the external magnetic field (red circular marks) shows 1.8 times higher coupling than that without an external magnetic field (blue

rectangular marks), and that electron scattering in the closed cone-tip (green triangular marks) reduces the coupling 0.8 times compared to the open-tip values (red circular marks). Furthermore, the coupling was degraded gradually by increasing the heating laser energy with keeping both pulse duration and spot diameter unchanged (1.8 ps and 50 μm), because a higher-intensity laser produces higher-temperature REB, as shown in Table 2 and this results in less guiding and less energy deposition. We anticipate that the pulse duration should be extended to sustain this efficient coupling for higher laser energy.

As shown in Fig. 2, Cu-$K_\alpha$ X-ray yield produced during the compression process (green dotted line) was negligibly weak compared with those produced during the heating process (red solid and black dashed lines) even with application of the external magnetic field. This result means that Cu-$K_\alpha$ enhancement is not due to an improved confinement of thermal electrons in the compressed plasma but to the guiding of the REB.

Figure 6 shows two-dimensional Cu-$K_\alpha$ emission profiles and density profiles of the pre-compressed core at two different

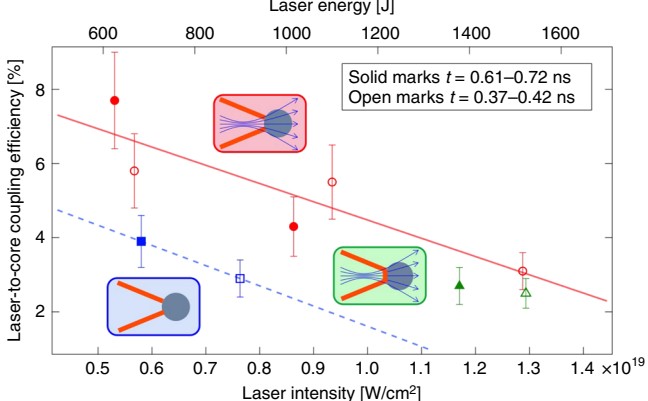

**Fig. 5** Dependence of laser-to-core energy coupling on heating laser intensity and energy. The blue rectangular, green triangle, and red circle marks represent laser-to-core coupling efficiencies obtained with the following conditions: no application of an external magnetic field with the open-tip cone, application of an external magnetic field with the closed-tip cone, and application of an external magnetic field with the open-tip cone, respectively. The error bars show the uncertainty of the experimental measurement described in the text. Solid and open marks represent the couplings of two injection-timing groups $t = 0.61$–$0.72$ ns and $t = 0.37$–$0.42$ ns. The error bars are the spectrometer uncertainty due to nonuniformity of the the the integral reflectance. The solid and dashed lines are fitted, as an eye guide, to the couplings with neglecting the injection timing difference

timings ($t = 0.40 \pm 0.03$ ns and $0.69 \pm 0.04$ ns), and also the comparison of Cu-$K_\alpha$ emission profiles with and without the external magnetic field application at the two different timings. In the Cu-$K_\alpha$ emission profiles, the numbers on the contour lines represent Cu-$K_\alpha$ emissivity normalized with the heating laser energy [$\times 10^9$ photons sr$^{-1}$ cm$^{-3}$ J$^{-1}$], which was calculated from the combination of the absolute Cu-$K_\alpha$ spectra and the two-dimensional Cu-$K_\alpha$ monochromatic images after applying an inverse Abel transformation to the line-integrated emission profiles assuming rotational symmetry of the emissivity along the cone axis.

At $t = 0.40 \pm 0.03$ ns, the converging shock wave was still traveling to the center of the ball, and the shock front is clearly observable in Fig. 6b. Strong Cu-$K_\alpha$ emission region locates behind the shock front in Fig. 6a and e. Externally applied magnetic field lines were accumulated in the downstream of the shock wave; therefore, REB was guided to the shock-compressed region along the field lines. This feature was less prominent without the external magnetic field as shown in Fig. 6f.

At $t = 0.69 \pm 0.04$ ns, the solid ball reached the maximum compression (Fig. 6d). The strong Cu-$K_\alpha$ emission spot appeared at 50-μm longitudinal distance, which was more than 100 μm away from the REB generation zone, namely the cone tip as shown in Fig. 6c and g. This Cu-$K_\alpha$ emission feature is the evidence of the long-distance REB guiding by the externally applied magnetic field as well as significant enhancement of the laser-to-core coupling. This strong emission spot disappeared, when the external magnetic field was not applied, as shown in Fig. 6h. Energy shift of the Cu-$K_\alpha$ X-ray due to ionization of Cu atoms in a hot core could be the reason why the Cu-$K_\alpha$ emission was weak in the core central region[40]. Note that the strong emission spot that appeared in Fig. 6c and g was found also in the other shot (ID 40556) performed with similar injection timing ($t = 0.61 \pm 0.02$ ns).

The laser-to-core coupling ($\eta$) can be simplified as a product of laser-to-REB energy conversion efficiency ($\eta_{REB}$), REB collision probability ($\eta_{coll}$), and energy deposition rate of REB in the core ($\eta_{dep}$)[41].

$$\eta = \eta_{REB} \cdot \eta_{col} \cdot \eta_{dep} \tag{1}$$

The previous experiments show that $\eta_{REB} = 0.4$[41] and $\eta_{coll} = 0.7$[29]. $\eta_{dep}$ was calculated by the simplified model[41] with the measured areal density and REB temperatures shown in Table 2.

$$\eta_{dep} = \frac{AT_{REB1}^2}{AT_{REB1}^2 + (1-A)T_{REB2}^2} \cdot \frac{\rho L}{0.6 T_{REB1}} + \frac{(1-A)T_{REB2}^2}{AT_{REB1}^2 + (1-A)T_{REB2}^2} \cdot \frac{\rho L}{0.6 T_{REB2}}. \tag{2}$$

As an approximated relation of Eq. (11) in ref.[42], $R_{REB}$ [g cm$^{-2}$] = $0.6 f_R T_{REB}$ [MeV] is used to calculate the $R_{REB}$ from

**Table 2 Summary of correlation factors used in the analysis**

| Shot ID | A | $T_{REB1}$ [MeV] | $T_{REB2}$ [MeV] | Correlation factor with Davis model [J photons$^{-1}$sr$^{-1}$] | Correlation factor with Hombourger model [J photons$^{-1}$sr$^{-1}$] | Correlation factor used in the analysis [J photons$^{-1}$sr$^{-1}$] |
|---|---|---|---|---|---|---|
| 40545 | 0.881 | 1.0 | 4.7 | $4.1 \times 10^{-11}$ | $5.3 \times 10^{-11}$ | $4.7 \times 10^{-11}$ |
| 40541 | 0.951 | 0.7 | 4.4 | $4.2 \times 10^{-11}$ | $5.5 \times 10^{-11}$ | $4.8 \times 10^{-11}$ |
| 40558 | 0.956 | 4.6 | 23.6 | $3.4 \times 10^{-11}$ | $4.5 \times 10^{-11}$ | $4.0 \times 10^{-11}$ |
| 40556 | 0.933 | 2.2 | 5.4 | $3.7 \times 10^{-11}$ | $4.9 \times 10^{-11}$ | $4.3 \times 10^{-11}$ |
| 40547 | 0.907 | 1.6 | 2.8 | $4.1 \times 10^{-11}$ | $5.4 \times 10^{-11}$ | $4.7 \times 10^{-11}$ |
| 40549 | 0.999 | 0.8 | 10 | $4.7 \times 10^{-11}$ | $6.0 \times 10^{-11}$ | $5.3 \times 10^{-11}$ |
| 40543 | 0.971 | 0.5 | 4.1 | $4.5 \times 10^{-11}$ | $5.8 \times 10^{-11}$ | $5.1 \times 10^{-11}$ |
| 40560 | 0.991 | 0.9 | 21.7 | $3.9 \times 10^{-11}$ | $5.2 \times 10^{-11}$ | $4.5 \times 10^{-11}$ |
| 40562 | 0.890 | 1.5 | 5.6 | $4.0 \times 10^{-11}$ | $5.2 \times 10^{-11}$ | $4.6 \times 10^{-11}$ |

The data are separated into three groups as Table 1. A, $T_{REB1}$, and $T_{REB2}$ are the intercept and the two slope-temperatures of the REB energy distribution, respectively.

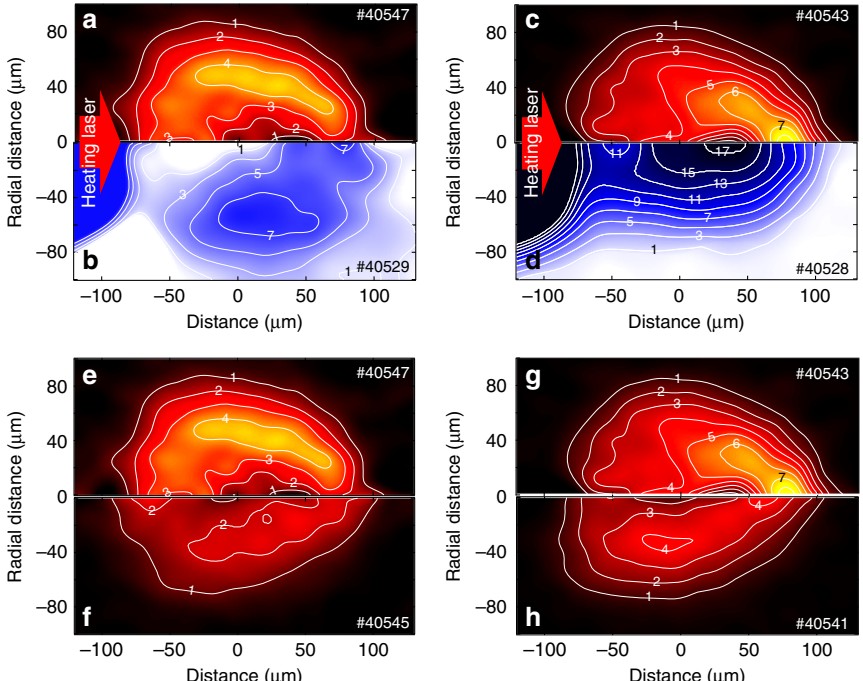

**Fig. 6** Two-dimensional emission and density profiles. Two-dimensional profiles of Cu-$K_\alpha$ emission (**a**, **c**, **e**, **g**) and mass density (**b**, **d**). These profiles were measured in the experiments with the application of the external magnetic field at $t = 0.40 \pm 0.03$ ns (**a**, **b**, **e**) and $t = 0.69 \pm 0.04$ ns (**c**, **d**, **g**). The numbers on the contour lines represent heating-laser energy-normalized Cu-$K_\alpha$ emissivity ($\times 10^9$ photons sr$^{-1}$ cm$^{-3}$ J$^{-1}$) and mass density (g cm$^{-3}$), respectively. Cu-$K_\alpha$ emission profiles are compared between those obtained with (**e**, **g**) and without (**f**, **h**) application of the external magnetic field at $t = 0.40 \pm 0.03$ ns and $t = 0.69 \pm 0.04$ ns. These images were obtained after applying an inverse Abel transformation to the line-integrated emission profile, assuming rotational symmetry of the core along the cone axis

the experimentally measurable parameter ($T_{REB}$), here, $f_R$ is an adjustable parameter, set to 1 in the standard model. Finally, the simple model yields $\eta = \eta_{REB} \cdot \eta_{coll} \cdot \eta_{dep} = 6.2\%$. This simple evaluation seems fairly consistent with the measured coupling ($7.7 \pm 1.2\%$), and the simple evaluation reveals that higher areal density core leads to higher laser-to-core coupling. An ultra-high-energy-density state could be efficiently created by the magnetized fast isochoric heating. Integrated REB transport simulation result is described in the Supplementary Discussion and Supplementary Fig. 1. The simulation result is fairly consistent with the experimental results.

We have demonstrated experimentally the magnetized fast isochoric heating scheme for creating an ultra-high-energy-density state as a potential path to the ignition spark formation. The enhancement of the laser-to-core coupling as well as the strong Cu-$K_\alpha$ emission spot located at 100 μm away from the cone tip are the evident features produced by guiding of the diverged REB with the externally applied magnetic field in the long-transport distance. An energy-density increment of the heated core is close to 1 Gbar, which corresponds to 50 J of the energy deposition in a 100-μm-diameter spherical volume. Plasma hydrodynamics, generation and transport of electron/ion beams, thermal conduction, and α-particle transport will be able to be controlled by the externally applied strong magnetic field. There is no doubt that laser–plasma experiments with strong magnetic fields contain a lot of unexplored physics; therefore, this research also stimulates spin-off sciences in the field of atomic physics, nuclear physics, and astrophysics which act to broaden inertial fusion sciences and high-energy-density sciences.

## Methods

**Derivation of the correlation factor between Cu-$K_\alpha$ photons and deposited REB energy**. The Solodov model[43] was used to calculate the stopping power $S(E,\rho)$ [J s$^{-1}$] of the REB in a core, where $E$ is electron kinetic energy. Both Davies[44] and

Hombourger[45] models were used to calculate the electron-impact K-shell ionization cross-section $\sigma_{K_\alpha}(E)$ [m$^{-2}$]. The ratio between the stopping power and K-shell ionization cross-section is the correlation factor ($C$ [J photons$^{-1}$ sr]). Note that the Hombourger model, which was used in the previous experiment[18], gives a 1.3 times higher correlation factor than the Davies model because $\sigma_{K_\alpha}(E)$ calculated with the Davies model is larger than that with the Hombourger one. The correlation factor $C$ is defined as follows:

$$C = \frac{\int_0^L \int_{10\,keV}^{1\,GeV} \varepsilon_{dep}(E,x)\,dEdx}{\int_0^L \int_{10\,keV}^{1\,GeV} P_{K_\alpha}(E,x)\,dEdx} \tag{3}$$

$$= \frac{\int_0^L \int_{10\,keV}^{1\,GeV} v(E,x)f(E,x)S(E,\rho)\,dEdx}{\int_0^L \int_{10\,keV}^{1\,GeV} v(E,x)f(E,x)n_{Cu}\sigma_{K_\alpha}(E)/4\pi\,dEdx} \tag{4}$$

where $\varepsilon_{dep}(E,x)$, $P_{K_\alpha}(E,x)$, $v(E,x)$, $f(E,x)$, and $n_{Cu}$ are the collisionally deposited energy by the REB to the dense core [J s$^{-1}$], the probability of Cu-$K_\alpha$ emission [photons s$^{-1}$ sr$^{-1}$], the relativistic electron velocity [m s$^{-1}$], the REB energy distribution, and the number density of Cu atoms in a core [m$^{-3}$], respectively. The initial REB energy distribution was a Boltzmann function as $f(E,0) = \exp(-E/T_{REB})$ at the generation zone ($x = 0$), where $T_{REB}$ is the slope temperature of the energy distribution. The distribution at x ($f(E,x)$) is calculated by considering the collisional slowing down during the transport.

Figure 7 shows the calculated correlation factor at different REB slope temperatures ($T_{REB}$) for a $\rho = 11.3$ g cm$^{-3}$ and $\rho L = 0.16$ g cm$^{-2}$ Cu(II) oleate plasma, which are equal to those observed at $t = +0.72$ ns. The correlation factor also depends weakly on the core density. Dependence of the correlation factor on $T_{REB}$ was considered in the coupling evaluation. The average values of the two correlation factors calculated with the two models were used in the data analysis.

Energy distributions of electrons escaped from the plasma into vacuum were measured using an electron energy analyzer positioned along the LFEX incident axis. Although the energy distribution of the so-called vacuum electrons is not exactly identical to that at the generation zone due to scattering, absorption, and reflection by the cone, core, and spontaneous electromagnetic fields, we found that the slope temperatures of the escaped electrons are close to those in the transport region estimated from Bremsstrahlung X-ray spectra[41]. Therefore, the energy distribution of the measured vacuum electrons was used in the correlation factor calculation. The energy distribution of the escaped electrons was fitted with a two-temperature Boltzmann distribution function as $f(E) = A \exp(-E/T_{REB1}) + (1-A) \exp(-E/T_{REB2})$, where $A$ and $E$ are the intercept and electron energy, and $T_{REB1} < T_{REB2}$. The correlation factor calculated for each shot is summarized in Table 2

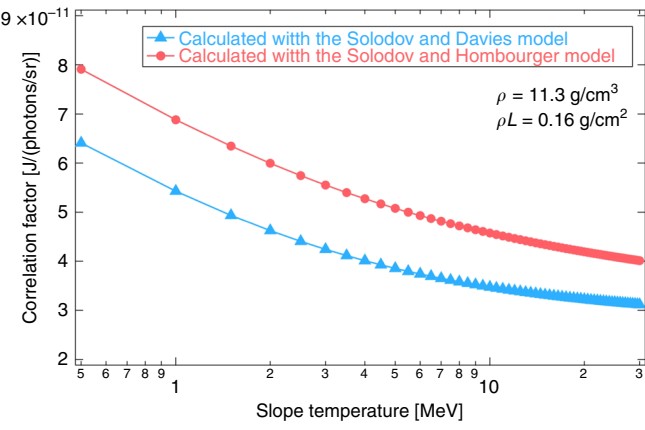

**Fig. 7** Correlation factor. Dependence of the correlation factor between the deposited energy (J) and Cu-$K_\alpha$ yield (photons/sr) on the REB slope temperature. The data were calculated using the Davies (blue circular marks) and Hombourger (red triangular marks) models of electron-impact K-shell ionization for a 11.3 g cm$^{-3}$ and 0.16 g cm$^{-2}$ Cu(II) oleate, which correspond to the average core density and areal density at $t = 0.72$ ns (maximum compression timing). The average values of the two correlation factors calculated were used in the analysis

along with other measured parameters. The error in the deposited energy was evaluated taking into account 16% of spatial nonuniformity of the integral reflectance of the HOPG.

**Code availability**. The computer codes used in the current study are accessible from the corresponding author upon reasonable request.

## Data availability

The datasets analyzed during the current study are available from the corresponding author upon reasonable request. In accordance with the guideline for research data storage at the Institute of Laser Engineering, Osaka University, all data are properly stored in the SEDNA database system.

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

## Acknowledgements

The authors thank the technical support staff of ILE and the Cyber Media Center at Osaka University for assistance with the laser operation, target fabrication, plasma diagnostics, and computer simulations. The authors greatly appreciate valuable discussions with Drs. J. Moody, B. Pollock, M. Tabak, W. Kruer, O. Landen, T. Ma, H. Chen, A. Kemp, D. Mariskal, and B. Remington (LLNL), Profs. M. Murakami and K. Shigemori and Drs. Y. Sakawa and T. Sano (ILE, OU), Dr. Iwamoto (NIFS), Dr. K. Kondo (QST), and especially Dr. S. Wilks (LLNL) also for his proofreading of the first draft. This work was supported by the Collaboration Research Program of the Institute of Laser Engineering at Osaka University and also the Collaboration Research Program  between the National Institute for Fusion Science and the Institute of Laser Engineering at Osaka University (NIFS12KUGK057, NIFS15KUGK087, NIFS17KUGK111, and NIFS18-KUGK118), and by the Japanese Ministry of Education, Science, Sports, and Culture through Grants-in-Aid, KAKENHI (Grant No. 24684044, 25630419, 15K17798, 15K21767, 15KK0163, 16K13918, 16H02245, and 17K05728), Bilateral Program for Supporting International Joint Research by JSPS, and Grants-in-Aid for Fellows by Japan Society for the Promotion of Science (Grant No. 14J06592, 15J00850, 15J00902, 15J02622, 17J07212, 18J01627, 18J11119, and 18J11354). The study also benefited from diagnostic support funded by the French state through research projects TERRE ANR-2011-BS04-014 (French National Agency for Research (ANR) and Competitiveness Cluster "Route des Lasers") and ARIEL (Regional Council of Aquitaine). M.B.-G. and J.J.S. acknowledge the financial support received from the French state and managed by ANR in the framework of the "Investments For the Future" program at IdEx Bordeaux—LAPHIA (ANR-10-IDEX-03-02), from COST Action MP1208 "Developing the Physics and the Scientific Community for Inertial Fusion" and from the EUROfusion Consortium and have received funding from the Euratom research and training programs 2014–2018 under grant agreement No. 633053. The views and opinions expressed herein do not necessarily reflect those of the European Commission.

## Author contributions

S. F. is the principal investigator who proposed and organized this study. The experiment was carried out by S.S., S.L., H.M., K.M.A., K.F.F.L., A.Y.A., S.K., Y.A.B., H.K., and A.S.Y. under the supervisions by H.S.A.W., A.M., A.Y.O., M.N., T.O., H.S.H., H.N.I., H.A., R.K., Y.A.R., and S.F.H.M. and A.S.U. developed the code to simulate magnetic field diffusion. S.S., S.L., T.J. and S.F. analyzed the X-ray spectra to calculate the coupling. S.S., S.L. and H.S.A.W. analyzed the Cu-$K_\alpha$ emission and backlight images. A.Y.A., A.Y.O., and H.N.I. measured the absolute reflectivity of the HOPG. Y.I., K.Y. and T.N. developed targets used in the experiment. Y.N., S.T., N.M. and J.K. contributed to improve intensity contrast of the LFEX laser pulses. T.J., M.H., A.S.U., T.S., N.I., H.S.A.K., K.M.I., H.N.A. and Y.S. performed computer simulation and theoretical analysis for determining the experimental conditions. M.B.-G. and J.J.S. contributed to the design of the integrated MFI experiment based on their experimental results obtained in the LULI2000 facility. All authors contributed to the discussion of the results.

## Additional information

**Competing interests:** The authors declare no competing interests.

