## [Peer Review File · Nature Communications]

Reviewers' comments:

Reviewer #1 (Remarks to the Author):

This article presents experimental results on improved coupling of a relativistic electron beam to a compressed solid-density target. The improvement is due to an imposed, laser-drive, kilo-Tesla magnetic field, as well as using an initially solid target, instead of a thin shell as is commonly used for inertial fusion. The results are promising and seem valid. But I do not think this rises to the level of Nature Communications.

The paper combines ideas that have already been proposed, and is therefore not really breaking new ground. Also the improved coupling isn't that significant, in fact close to results the authors cite from OMEGA (albeit with higher $\rho \cdot r$ and reduced pre-plasma). The solid-density target (as opposed to a thin shell) does not appear on a path that scales to ICF ignition, so these results do not improve the prospects of fusion in an obvious or immediate way.

This paper is probably appropriate for Physical Review Letters, or possibly Nature Physics. There are very few plasma-physics papers in Nature Communications, maybe several to ten a year. Is this one of the 10 best plasma-physics results of 2018? In my opinion it is not.

Reviewer #2 (Remarks to the Author):

The results presented in this paper are potentially of interest to the fusion community, however they are based on a very limited number of experimental data points, which means that it is difficult to have confidence in the claimed results. At this point I cannot pass judgement on their claimed findings without additional information.

The authors' claims principally rest on the results presented in figures 2 and 3. I will discuss these in detail.

Figure 2:

- within the margins of the error bars the green points are indistinguishable from the red, so I am ignoring these.
- Differentiation of the blue points from the red, relies on the error bars of the blue points being less than that of the red. Can the authors justify these reduced error bars?
- This plot is comprised of points which have huge differences in the timing of the heating beam with respect to one another. It is unclear that it is justified making a direct comparison of these points. Can the authors justify this?

Figure 3:

- This plot is a comparison of shots with and without magnetic fields, however the shots the authors have chosen to compare have very different heating laser energies, so it isn't at all clear whether their claims can be justified. I would like to see these plots re-made with shots of similar energy (e.g. 40541 vs 40543) and also with the K alpha emission normalised to the incident laser energy.

At this point I will not detail other more minor issues I have with the paper.

Reviewer #3 (Remarks to the Author):

The manuscript describes an experiment on possible enhancement of fast ignition by applying a strong magnetic field to guide and to confine fast electrons. Although there seems to be a positive effect, the interpretation and modeling of the results are far from being sufficient for publishing in Nature. At best, the manuscript can be qualified as an internal progress report to serve as a motivation for further investigation.

1. The authors rely on previous works on magnetic field generation by the “Capacitor Coil” method. Since the criticism of the previous works is outside of the scope of this review, I will just mention a few points in this particular manuscript that seem doubtful:
 - From the geometry of the “coil”, it seems roughly equivalent, including the lead wires, to a 2mmx0.5mm loop, that is having a cross-section of $A \sim 1\text{mm}^2$. Driving a current of 250kA would create a field of $B \sim 600\text{T}$ inside the coil. Assuming a field rise time of $\tau \sim 0.5\text{ns}$, that would require a voltage of about $AB/\tau \approx 1.2\text{MV}$ applied to the coil which is about two orders of magnitude higher than the measured fast electron temperature!
 - The authors speculate that using three lasers will magnify the magnetic field by a factor of $3^{1/2}$. However, this implies that the magnetic field energy scales proportionally to the laser energy - the scaling that has not been established in the manuscript.
2. Details on the field penetration analysis into the cone and the cone tip foil are very sketchy. In particular, a time scale of $\mu_0\sigma L^2 \approx 2.5\text{ns}$ is mentioned. However, the field penetration time scale really depends on the specifics of the geometry. For example, for a cylinder (which is topologically equivalent to the cone) with a radius a and a wall thickness δ , the penetration time scale of an axial magnetic field is $\mu_0\sigma a\delta/2 \approx 18\text{ns}$ for $a = 100\mu\text{m}$ and $\delta = 7\mu\text{m}$, which is a factor of 7 longer than 2.5ns. Penetration into the cone tip, correctly analyzed, is also longer by almost the same factor. Given that the field lasers are peaked only at about 2 ns before the compression lasers, the field would not have any time to penetrate through the cone wall and the tip.
3. Fast electrons guiding and confinement has not been properly analyzed. The authors just briefly mention that “These strengths are high enough to guide the REB”. I think this moment is absolutely essential to the analysis of the results and it definitely deserves more than just a single sentence. Just an example related to item #2 - if the field is low at the cone tip due to the penetration effect and it is high away from the tip then the electrons emerging at the tip would have to climb up a strong magnetic hill and it is not clear what fraction of them would reach the target.
4. Finally, it is not clear from the analysis whether the observed enhancement of the K_alpha radiation is due to the fast electrons or just to an improved confinement of thermal electrons in the compressed plasma. An experiment that could verify it would be to make shots with an applied magnetic field but not firing the short pulse REB lasers.

Authors' response to Reviewer1

The authors appreciate the reviewer's time and effort for reviewing our manuscript. Your comments, suggestions, and criticisms helped us greatly to improve the quality of this manuscript. The previous manuscript was written in a Letter style of an another Nature journal. We reformatted thoroughly the manuscript to be adequate to Nature Communications also with consideration of your inputs. We hope that the revised manuscript satisfies the standard to warrant publication of our research in Nature Communications.

Comment 1:

This article presents experimental results on improved coupling of a relativistic electron beam to a compressed solid-density target. The improvement is due to an imposed, laser-drive, kilo-Tesla magnetic field, as well as using an initially solid target, instead of a thin shell as is commonly used for inertial fusion. The results are promising and seem valid. But I do not think this rises to the level of Nature Communications. This paper is probably appropriate for Physical Review Letters, or possibly Nature Physics. There are very few plasma-physics papers in Nature Communications, maybe several to ten a year. Is this one of the 10 best plasma-physics results of 2018? In my opinion it is not.

Response from the authors:

We guess that the reviewer misunderstood *Nature Communications* journal as Communications in *Nature* journal because the reviewer recommends this paper to be possibly appropriate for *Nature Physics* instead of *Nature Communications*. While *Nature Physics* requests that published researches have the highest impact within the disciplines, *Nature Communications* is committed to publishing important advances of significance to specialists within each field. We think that our manuscript satisfies, at least, the standard of *Nature Communications* even if we accept the reviewer's evaluation for our paper.

Comment 2:

The paper combines ideas that have already been proposed, and is therefore not really breaking new ground. Also the improved coupling isn't that significant, in fact close to results the authors cite from OMEGA (albeit with higher $\rho \cdot r$ and reduced pre-plasma). The solid-density target (as opposed to a thin shell) does not appear on a path that scales to ICF ignition, so these results do not improve the prospects of fusion in an obvious or immediate way.

Response from the authors:

We disagree with this reviewer's comment. It needs long-term continuing efforts to realize the proposed ideas. We believe that breaking new ground is really happen only by preforming experiments not just by proposing ideas. While 7% of the coupling was obtained using a 0.3 g/cm² core in OMEGA facility, we achieved 7.7% coupling at core area density 0.1 g/cm². This coupling per unit area density is three times more efficient than the OMEGA value, which justifies clearly the great advantage of the magnetized fast isochoric heating scheme.

The solid ball target had not been considered as an ignition target, however, we are now investigating its possibility of forming an ignition scale core. The graph shows a temporal evolution of area density calculated with a one-dimensional hydrodynamic simulation code for a solid DT filled in a 2-mm-diameter and 25- μ m-thick plastic shell compressed by 0.35 μ m and 300 kJ laser beams. In the fast ignition scheme, the solid ball target can be an ignition target because the hot spark is produced separately by the external energy injection.

Authors' response to Reviewer 2

The authors appreciate the reviewer's time and effort for reviewing our manuscript. Your comments, suggestions, and criticisms helped us greatly to improve the quality of this manuscript. The previous manuscript was written in a Letter style of an another Nature journal. We reformatted thoroughly the manuscript to be adequate to Nature Communications also with consideration of your inputs. We hope that the revised manuscript satisfies the standard to warrant publication of our research in Nature Communications.

Comment 1:

Figure 2 (***This is changed to Figure 6 in this revised manuscript***):

1-A) within the margins of the error bars the green points are indistinguishable from the red, so I am ignoring these.

1-B) Differentiation of the blue points from the red, relies on the error bars of the blue points being less than that of the red. Can the authors justify these reduced error bars?

1-C) This plot is comprised of points which have huge differences in the timing of the heating beam with respect to one another. It is unclear that it is justified making a direct comparison of these points. Can the authors justify this?

Response from the authors:

In the first, we must explain the source of the error of the coupling. We used a planar highly oriented pyrolytic graphite (HOPG) in the X-ray spectrometer. The absolute integral reflectance of the HOPG was measured using an X-ray diffractometer. The HOPG has $\pm 16\%$ of spatial non-uniformity of its reflectance. It is not possible to identify the exact area of the HOPG that diffracts the detected Cu- $K\alpha$ X-rays, therefore the reflectance non-uniformity causes $\pm 16\%$ of uncertain in the evaluation of absolute deposit energy.

It is important to refer here that the HOPG alignment in respect to the targets was not changed during these measurements, and the detected $K\alpha$ photons were always reflected from the same area in the HOPG. This means that true couplings do not scatter randomly within the error bars but locates at certain points keeping ratios between another points within the error bars.

The above sentences have been written in the revised manuscript.

Figure 6 of the revised manuscript. Dependence of laser-to-core energy coupling on heating laser intensity (bottom axis) and energy (top axis). The blue rectangular, green triangle and red circle marks represent laser-to-core coupling efficiencies obtained with the following conditions; no application of external-magnetic-field with open-tip cone, application of external magnetic field with closed-tip cone, and application of external magnetic field with open-tip cone, respectively. Solid and open marks represent the couplings of two injection timing groups $t = 0.61 - 0.72$ ns and $t = 0.37 - 0.42$ ns. The solid and dashed lines are fitted, as an eye guide, to the couplings with neglecting the injection timing difference.

1-A) The green points are distinguishable from the red ones as discussed above.

1-B) The errors correspond to $\pm 16\%$ of the evaluated coupling due to the HOPG non-uniformity, therefore, the smaller coupling has the smaller error value.

1-C) The comment is completely correct. The solid and dashed lines were fitted, as an eye guide, to the couplings neglecting the injection timing difference. We have modified the figure by using the solid and open marks to make the injection time difference easily distinguishable.

Comment 2:

Figure 3 (***This is changed to Figure 7 in the revised manuscript***):

This plot is a comparison of shots with and without magnetic fields, however the shots the authors have chosen to compare have very different heating laser energies, so it isn't at all clear whether their claims can be justified. I would like to see these plots re-made with shots of similar energy (e.g. 40541 vs 40543) and also with the K alpha emission normalized to the incident laser energy.

Response from the authors:

We appreciate for this suggestion. We have redraw the panels with 40541 and 40543 shots using absolute $K\alpha$ emissivity normalized with the heating energy in the unit of $\times 10^9$ photons/str/cm³/J. The differences can be seen more clearly between these two shots compared to the previous plots. We have added the sentence as " Note that the strong emission spot appeared in Fig. 7 (c, g) can be seen also in the other shot (ID 40556) performed with close injection timing ($t = 0.61 \pm 0.02$ ns)."

Figure 7 of the revised manuscript. Two dimensional profiles of Cu- $K\alpha$ emission (a,c,e,g) and mass density (b,d) measured in the experiments with the application of the external magnetic field at $t = 0.40 \pm 0.03$ ns (a, b, e) and $t = 0.69 \pm 0.04$ ns (c,d,g). The numbers on the contour lines represent heating-laser-energy-normalized Cu- $K\alpha$ - emissivity ($\times 10^9$ photons/str/cm³/J) and mass density (g/cm³), respectively. Cu- $K\alpha$ emission profiles are compared between those obtained with (e,g) and without (f,h) application of the external magnetic field at two different injection timings. These images were obtained after applying an inverse Abel transformation to the line-integrated emission profile, assuming rotational symmetry of the core along the cone axis.

Authors' response to Reviewer 3

The authors appreciate the reviewer's time and effort for reviewing our manuscript. Your comments, suggestions, and criticisms helped us greatly to improve the quality of this manuscript. The previous manuscript was written in a Letter style of an another Nature journal. We reformatted thoroughly the manuscript to be adequate to Nature Communications also with consideration of your inputs. We hope that the revised manuscript satisfies the standard to warrant publication of our research in Nature Communications.

Comment 1:

From the geometry of the “coil”, it seems roughly equivalent, including the lead wires, to a 2mm x 0.5mm loop, that is having a cross-section of $A \sim 1\text{mm}^2$. Driving a current of 250kA would create a field of $B \sim 600\text{T}$ inside the coil. Assuming a field rise time of $\tau \sim 0.5\text{ns}$, that would require a voltage of about $AB/\tau \approx 1.2\text{MV}$ applied to the coil which is about two orders of magnitude higher than the measured fast electron temperature!

Response from the authors:

We recognized the problem what the reviewer pointed out. We have not yet been able to explain completely the mechanism, of which the non-thermal electron stream flows against such a strong electric field between the capacitor plates. Besides, V.T.Tikhonchuk *et al.* [Phys. Rev. E, Phys. Rev. E 96, 023202 (2017)] explained that the intense coil currents stem from the ion expansion from the laser irradiated zone: ions fill the volume between the target's disks in about 100ps, neutralizing the space charge and flattening the potential. The target then works like a laser-driven diode.

Comment 2:

The authors speculate that using three lasers will magnify the magnetic field by a factor of $3^{1/2}$. However, this implies that the magnetic field energy scales proportionally to the laser energy - the scaling that has not been established in the manuscript.

Response from the authors:

We remove the scaling from the revised manuscript. We revised the paragraph as " A current of 250 kA was generated with a capacitor-coil target driven by one GEKKO-XII beam as measured using proton radiography [12]. The current of 250 kA was used in the following analysis, although three GEKKO-XII beams were used in this integrated experiment."

Comment3:

Details on the field penetration analysis into the cone and the cone tip foil are very sketchy. In particular, a time scale of $\mu_0\sigma L/2 \approx 2.5ns$ is mentioned. However, the field penetration time scale really depends on the specifics of the geometry. For example, for a cylinder (which is topologically equivalent to the cone) with a radius a and a wall thickness δ , the penetration time scale of an axial magnetic field is $\mu_0\sigma a\delta/2 \approx 18ns$ for $a = 100\mu m$ and $\delta = 7\mu m$, which is a factor of 7 longer than 2.5ns. Penetration into the cone tip, correctly analyzed, is also longer by almost the same factor. Given that the field lasers are peaked only at about 2 ns before the compression lasers, the field would not have any time to penetrate through the cone wall and the tip.

Response from the authors:

We appreciate for introducing us the formula. We have rewritten the introduction of the magnetic field diffusion based on the cylindrical geometry that the reviewer pointed out.

In addition, we have developed an electro-magneto dynamics simulation code with consideration of the inductive heating and the temperature dependence of conductivity to calculate the spatial and temporal distribution of magnetic field in the cone-ball target. The magnetic field strengths at the tip is estimated to be 335 T by this calculation. This discussion has been written in the revised manuscript.

Details of the computation are written in H. Morita *et al.*, arXiv:1804.10410 (2018).

Figure 3 of the revised manuscript. Two-dimensional profile of the magnetic field generated with a coil, in which 250 kA of current flows at the field peak timing. The magnetic field diffusion was calculated using (a) the constant electrical

conductivities [$\sigma = 4 \times 10^7$ S/m] and (b) with consideration of temporal changes in temperature and conductivity of the gold cone due to the inductive heating.

Comment4:

Fast electrons guiding and confinement has not been properly analyzed. The authors just briefly mention that “These strengths are high enough to guide the REB”. I think this moment is absolutely essential to the analysis of the results and it definitely deserves more than just a single sentence. Just an example related to item #2 - if the field is low at the cone tip due to the penetration effect and it is high away from the tip then the electrons emerging at the tip would have to climb up a strong magnetic hill and it is not clear what fraction of them would reach the target.

Response from the authors:

We have added the discussion in the revised manuscript as "The magnetic field strengths at the ball center is $B_0 = 225$ T in Fig. 3 (b). The ball radius at the maximum compression timing was about $r_{\max} = 50$ μm as shown in Fig. 4 (b). The magnetic field strength at the maximum compression timing (B_{\max}) can be estimated with $B_{\max} = B_0(r_0/r_{\max})^{2(1-1/\text{Re}_m)}$, here $r_0 = 125$ μm and $\text{Re}_m \sim 2$ are initial radius of the ball and magnetic Reynolds number, respectively, as $B_{\max} = 560$ T. Therefore, the mirror ratio along the REB path, ratio of magnetic field strengths at the peak and the REB generation zone, is $R_m = 560/335 = 1.7$. This is small enough to guide the REB efficiently to the core in this system without significant losses caused by the mirror effect as discussed in Ref. [29]".

Comment5:

Finally, it is not clear from the analysis whether the observed enhancement of the K_{α} radiation is due to the fast electrons or just to an improved confinement of thermal electrons in the compressed plasma. An experiment that could verify it would be to make shots with an applied magnetic field but not firing the short pulse REB lasers.

Response from the authors:

As shown in Fig. 2, Cu- K_{α} X-ray yield produced during the compression process (green dotted line) was negligibly weak compared to those produced with the heating lasers (red solid and black dashed lines) even with application of the external magnetic field. This result means that Cu- K_{α} enhancement is not due to an improved confinement of thermal electrons in the compressed plasma but to the guiding of the fast electrons. This sentence has been written in the revised manuscript.

Figure 2 of the revised manuscript. An example of Cu- $K\alpha$ spectra peaked at 8.05 keV. Red solid, black dashed, and green dotted lines are, respectively, spectra obtained by heating with application of external magnetic field, heating without the application, and only fuel compression with application of external magnetic field.

Reviewers' comments:

Reviewer #1 (Remarks to the Author):

I thank the authors for clarifying the role of the journal Nature Communications. I almost never read any Nature journals unless someone points me to an article, and it's almost always Nature Physics. I was unaware of the massive number of journals now published under the Nature "brand." I read Physics of Plasmas and Physical Review Letters regularly, everything else I mostly ignore.

Since I never read Nature Comm., I don't have a good sense of the caliber of paper that gets accepted. I would say the work is of the level that Physical Review Letters publishes. So, I leave it to the editors to decide if the point of their journal is to publish at (or below) PRL level, or above it.

I appreciate the revisions the authors have made. I now think the paper should be published, provided it meets the editors' standards vs. PRL.

A few minor comments:

* I'd replace 'area density' with 'areal density', which is more standard.

* In Table 1, "Davis" should be "Davies".

Reviewer #2 (Remarks to the Author):

After the significant changes made to this paper, I am now happy to recommend it for publication.

Reviewer #3 (Remarks to the Author):

Review of the authors rebuttal:

Response from the authors:

We recognized the problem what the reviewer pointed out. We have not yet been able to explain completely the mechanism, of which the non-thermal electron stream flows against such a strong electric field between the capacitor plates. Besides, V.T.Tikhonchuk et al. [Phys. Rev. E, Phys. Rev. E 96, 023202 (2017)] explained that the intense coil currents stem from the ion expansion from the laser irradiated zone: ions fill the volume between the target's disks in about 100ps, neutralizing the space charge and flattening the potential. The target then works like a laser-driven diode.

Reviewer:

So, basically the authors are saying that they do not know what current is generated and they do not exactly understand its mechanism. There are many models of this phenomenon and without direct and firm measurements, the magnitude of the coil current still remains unknown.

Response from the authors:

We remove the scaling from the revised manuscript. We revised the paragraph as " A current of 250 kA was generated with a capacitor-coil target driven by one GEKKO-XII beam as measured using proton radiography [12]. The current of 250 kA was used in the following analysis, although three GEKKO-XII beams were used in this integrated experiment."

Reviewer:

Two things I would like to mention here. First, as I said before, the figure of the coil current of 250 kA remains questionable. Second, the statement that this number is used in the following analyses is not what is presented in the paper. For instance, the revised Bfield map in Fig. 3 looks identical to that in the original paper. In addition, it unclear whether all three Gekko beams (as in the original paper) are needed for the effect or just a single beam (as in the revised version) is enough.

Response from the authors:

We appreciate for introducing us the formula. We have rewritten the introduction of the magnetic field diffusion based on the cylindrical geometry that the reviewer pointed out. In addition, we have developed an electro-magneto dynamics simulation code with consideration of the inductive heating and the temperature dependence of conductivity to calculate the spatial and temporal distribution of magnetic field in the cone-ball target. The magnetic field strengths at the tip is estimated to be 335 T by this calculation. This discussion has been written in the revised manuscript. Details of the computation are written in H. Morita et al., arXiv:1804.10410 (2018).

Reviewer:

Again, the revised Fig. 3 is identical to the original one so it is hard to judge what changes have been introduced and to what effect. The authors reference an unreviewed paper in arXiv so it is impossible to ascertain its value.

Response from the authors:

We have added the discussion in the revised manuscript as "The magnetic field strengths at the ball center is $B_0 = 225$ T in Fig. 3 (b). The ball radius at the maximum compression timing was about $r_{max} = 50$ μm as shown in Fig. 4 (b). The magnetic field strength at the maximum compression timing (B_{max}) can be estimated with $B_{max} = B_0(r_0/r_{max})^2(1-1/Re_m)$, here $r_0 = 125$ μm and $Re_m \sim 2$ are initial radius of the ball and magnetic Reynolds number, respectively, as $B_{max} = 560$ T. Therefore, the mirror ratio along the REB path, ratio of magnetic field strengths at the peak and the REB generation zone, is $R_m = 560/335 = 1.7$. This is small enough to guide the REB efficiently to the core in this system without significant losses caused by the mirror effect as discussed in Ref. [29]".

Reviewer:

Again, without a proper validation of the magnetic field measurement and field profile calculation, and it is one of the central points of the paper, it is impossible to ascertain the accuracy of these statements.

Authors' response to Reviewer 1

The authors appreciate the reviewer's time and effort for reviewing carefully our manuscript.

Comment 1 :

- * I'd replace 'area density' with 'areal density', which is more standard.
- * In Table 1, "Davis" should be "Davies".

Response from the authors:

The issues mentioned above have been corrected in the revised manuscript.

Authors' response to Reviewer 3

The authors appreciate the reviewer's time and effort for reviewing carefully our manuscript. Your comments, suggestions, and criticisms helped us greatly to improve the quality of this manuscript. We hope that the revised manuscript and the supplemental material satisfy the standard to warrant publication of our research in Nature Communications.

Comment 1 :

The authors are saying that they do not know what current is generated and they do not exactly understand its mechanism. There are many models of this phenomenon and without direct and firm measurements, the magnitude of the coil current still remains unknown. As I said before, the figure of the coil current of 250 kA remains questionable.

Without a proper validation of the magnetic field measurement and field profile calculation, and it is one of the central points of the paper, it is impossible to ascertain the accuracy of these statements.

Response from the authors:

Here we will explain two facts to strengthen our conclusion. One is a summary of the magnetic field produced by the laser-driven scheme and the other is the integrated REB transport simulation with changing applied magnetic field strength. There are written in the Supplementary Information.

We stress that 250 kA is not questionable based on the previous experimental results. Table R1 summarizes previous experimental results obtained with kilo-Joule class laser facilities. Current flows in the coils were evaluated from the measured magnetic field strengths with resistances and inductances that were calculated for the initial coil geometries. 200 kA-level currents were obtained with except for Ref. 27 [L. Gao et al., Phys. Plasmas 23, 043106 (2016).]. In Ref. 27, a plastic spacer was inserted between the capacitor-plates, a current may flow in not only the coil but also the spacer surface. This target design is completely different from the other ones used in Refs. 24, 25, 26 and 28.

In addition, the most important result of this manuscript is not affected by the exact value of the current and also understanding of the current generation mechanism. The enhancement of the laser-to-core coupling is the consequence of REB guiding by the

applied magnetic field. The magnetic field strength, which was measured directly in the experiments, is important for the REB guiding.

Table. R1 Summary of magnetic field strength measured on kilo-Joule class laser facilities.

Facility /Country	Laser Energy (J)	Irradiance ($W \cdot cm^{-2} \cdot \mu m^2$)	Pulse duration (ns) / shape	Coil diameter (mm) / Material	Methods	B-field strength @center (T)
LULI (France) [Ref.25]	500	1×10^{17}	1.0 / Square	0.5 / Ni	• Proton • B-dot • Faraday	600 +/- 10 (300 kA)
GEKKO-XII (Japan) [Ref.24]	540	2×10^{16}	1.3 / Gaussian	0.5 / Ni	• Proton • B-dot	610 +/- 30 (250 kA)
OMEGA EP (U.S.) [Ref.26]	1250	2×10^{15}	1.0 / Square	0.3 / Cu w / spacer	• Proton	50 (22 kA)
OMEGA EP (U.S.) [Ref.27]	750	5×10^{14}	0.75 / Square	0.5 / Au 10 μ m-CH coated	• Proton • Faraday	210 +/- 35 (180 kA)
Shengguang-II (China) [Ref. 28]	1970	7×10^{14}	1.0 / Square	1.16 / Cu	• Proton • B-dot	205 (200 kA)

The integrated REB transport simulation was performed by coupling two-dimensional hybrid code [T. Johzaki *et al.*, Phys. Plasmas **16** 062706 (2009)] and two-dimensional radiative MHD code [H. Nagatomo *et al.*, Phys. Plasmas **14** 056303 (2007)].

Two-dimensional radiative MHD code calculated density and magnetic field profiles of a compressed solid ball attached to a gold cone. The initial magnetic field profile is shown in Fig. 3 (b) of the revised manuscript. The profiles at the maximum compression timing are shown in Figure R1 (a) and (b), respectively. The total energy and the wavelength of the compression laser were 1.5 kJ and 0.53 μ m, respectively. The pulse shape was a Gaussian with 1.3 ns of FWHM.

The REB transport simulation was performed with the profiles shown in Figure R1 (a) and (b). The REB was injected at $z = 65 \mu$ m (dashed white line). The half divergence angle of REB was 45 degrees. Temporal shape and spatial profile of the injected REB were a Gaussian with 1 ps duration (FWHM) and the super Gaussian with 30 μ m radius (FWHM), respectively. The peak intensity of REB was 7×10^{18} W/cm². The energy

distribution of REB was $f(E) \propto 0.76 \exp(-E/0.9[\text{MeV}]) + 0.24 \exp(-E/5[\text{MeV}])$ from [Y. Arikawa *et al.*, “Optimization of hot electron spectra by using plasma mirror for fast ignition” presented at IFSA2015, Seattle NE USA, Sept. 2015]. This distribution was obtained by coupling high energy x-ray spectrometer and electron energy spectrometer.

The multiplication factor of the REB-to-core energy coupling was calculated by changing initial magnetic field strength at the coil center as shown in Figure R1 (c).

Figure R1. Enhancement of REB-to-core energy coupling was calculated with the integrated simulation by coupling two-dimensional radiative MHD code and two-dimensional Fokker-Planck code. Spatial profiles of (a) density and (b) magnetic field at the maximum compression timing that were calculated with two-dimensional radiative MHD simulation code. (c) Dependence of multiplication factor of REB-to-core energy coupling on the initial magnetic field strength at the coil center that was calculated with two-dimensional Fokker-Planck code.

The multiplication factor is the ratio between the couplings calculated with and without application of the external magnetic field. The blue hatching indicates the range of the experimentally obtained multiplication factor including errors that is the ratio between the red point (ID 40543) and blue square (ID 40541). The calculated factor for above 300 T at the coil center is in the experimental range, which corresponds to 120 kA in the coil. This simulation result also supports the generation of several hundred Tesla magnetic field. The magnetization of relativistic electrons and magnetic mirror effect [30] account for the factor reduction shown in Fig. R1 (c) with increasing applied magnetic field strength.

Comment 2 :

The statement that this number is used in the following analyses is not what is presented in the paper. For instance, the revised B field map in Fig. 3 looks identical to that in the original paper. Again, the revised Fig. 3 is identical to the original one so it is hard to judge what changes have been introduced and to what effect. The authors reference an unreviewed paper in arXiv so it is impossible to ascertain its value.

Response from the authors:

We deeply apologize for our mistake made in the first manuscript. All the magnetic field profiles shown in the first, second, and third (this) manuscripts were calculated with 250 kA, not 430 kA that was written in the first manuscript. Therefore, Fig. S4 (a) of the first manuscript (calculated with a constant conductivity [$\sigma = 4 \times 10^7$ S/m]) are same with Fig. 3 (b) of the second manuscript.

The model used in the second and this manuscripts is completely different from that in the first one as written in the second and this manuscript. A temperature-dependent electrical conductivity model was used in Fig. 3 (b) of the second and this manuscripts, while a constant electrical conductivity [$\sigma = 2 \times 10^6$ S/m] was used in the first manuscript. However, it is difficult to judge the difference between them as the reviewer pointed out, because we showed the profiles only at the peak field timing.

Two-dimensional profiles of the magnetic field at three different timings are shown in Fig. R2. The upper figures were calculated with the constant electrical conductivities [$\sigma = 2 \times 10^6$ S/m], the lower figures were calculated with consideration of temporal changes in temperature and conductivity of a gold cone due to inductive heating. Difference of diffusion process is clearly seen.

The electro-magnetic dynamics code used in this simulation is based on Maxwell solver using a finite-difference time-domain (FDTD) method coupled with a conductivity depending on material temperature.

Figure R2 Two-dimensional profiles of the magnetic field at three different timings calculated with the σ constant conductivity (upper) and with consideration of temporal changes in temperature and conductivity of gold due to inductive heating (lower).

Comment 3 :

In addition, it unclear whether all three Gekko beams (as in the original paper) are needed for the effect or just a single beam (as in the revised version) is enough.

Response from the authors:

Three GEKKO beams were used for the magnetic field generation in this experiment. We have never performed the integrated experiment with a single GEKKO beam for the magnetic field generation. For the effect on the magnetic field diffusion, a single GEKKO beam (600 T, 250 kA) may be enough because 250 kA is used in our simulation shown in the Fig. R2. According to the integrated transport simulation, a single GEKKO beam seems to be enough to enhance the coupling, however, experimental confirmations have not been performed yet.

REVIEWERS' COMMENTS:

Reviewer #3 (Remarks to the Author):

It appears now that the authors have done significant amount of work revising the manuscript and responding to my comments and questions.

I do not have any other objections and I would like to recommend this manuscript for publications.